# Multimodal instruction with AI-generated images for noun retention: Exploring semantic scene and materiality effects

Gaojie Ye[1], Shibo Yan[2,3]*

1 Public Course Teaching Department, Anhui Vocational College of Defense Technology, Lu'an, China,
2 Department Of Information Technology, Anhui Vocational College of Defense Technology, Lu'an, China,
3 Faculty of Engineering, Science and Technology, Kuala Lumpur University of Science and Technology, Kuala Lumpur, Malaysia

* 242924740@s.klust.edu.my

## Abstract

This study explores the effectiveness of integrating multimodal instruction with artificial intelligence (AI)-generated visual content into English noun vocabulary instruction, as compared to text-only instruction. Rather than treating visual instruction as an end in itself, the approach leverages generative image technology to create contextually relevant stimuli that align with cognitive principles of memory formation. A controlled experiment (text-only vs. text+AI-generated images) was conducted with 40 English learners recruited from China. Participants completed immediate and delayed recall tests, definition selection, image-to-word matching (available only in the multimodal condition), and semantic rating tasks. Results revealed that the multimodal group significantly outperformed the text-only group across all measures, with large effect sizes for memory retention and semantic understanding. However, the study design does not allow us to attribute this advantage to the AI-generated nature of the images, as no condition with traditional images was included. These findings indicate that multimodal presentation can support durable and meaningful vocabulary learning when visual materials are designed to reflect perceptual and contextual features that facilitate memory. The study highlights the pedagogical potential of combining multimodal materials with memory-informed instructional design in language education.

## Introduction

Vocabulary acquisition is a foundational component of second language learning, particularly for learners aiming to develop essential communicative abilities [1]. Whether they can build a solid vocabulary base is closely tied to their capacity for basic comprehension of the language and functional interaction in real-world contexts. Compared with verbs, adjectives, Adverbials and other word classes, nouns play a central role in semantic understanding. They not only account for

**Data availability statement:** The data underlying the results presented in the study are available from Protocols.io (dx.doi.org/10.17504/protocols.io.j8nlk1bx6g5r/v2).

**Funding:** The author(s) received no specific funding for this work.

**Competing interests:** The authors have declared that no competing interests exist.

approximately 30%−40%, but also core words to refer to people, things, places, concept, etc. As a result, nouns often serve as anchors for sentence meaning [2,3]. Research has shown that compared with other word types such as prepositions or conjunctions, nouns constitute a significant proportion of early vocabulary and are more readily associated with visual and contextual cues [4,5].

Despite for their importance, vocabulary instruction continues to face notable challenges in second language learning. That's especially true for nouns. For most learners, noun acquisition serves as the starting point of vocabulary development, which forms the basis for essential comprehension and everyday communication. Traditional instructional methods typically involve word lists, definitions, and example sentences, which often lack the semantic richness needed to support long-term retention [6,7]. For younger learners, static images are sometimes used to aid understanding, but these visuals may still fall short in conveying contextual depth. Although authentic materials such as newspapers, films, and websites can expose learners to real-world language use, they inevitably contain semantic noise—numerous irrelevant words and scenes that distract from the target vocabulary [6]. Furthermore, the existing multimodal approaches rarely include systematic scene construction or material-based encoding strategies, and few studies have examined how such visual-semantic features influence the retention of noun vocabulary [5].

Recently, some developments in artificial intelligence (AI)-generated content have introduced new opportunities for vocabulary instruction. Some are realized particularly through advanced image generation technologies [8,9]. The visual materials produced by these tools are not only semantically aligned with target vocabulary but also rich in contextual and perceptual detail. For example, when the participants learn a word like security, the tools generate images that may depict various relevant scenes, such as security personnel, signage, or office environments, which can reinforce the its meaning through different real-world contexts. Except for scene-level semantic meaning, these images can also vary in finer visual aspects such as material textures, lighting conditions, and viewing angles. This combination of contextual diversity and visual specificity helps create a more engaging and cognitively stimulating learning experience. By linking the form, meaning and environment of the word, such tailored materials have the potential to support stronger memory encoding and help the learners get more durable vocabulary retention. This approach is consistent with cognitive perspectives, which emphasize the role of multimodal input in language learning and memory development [4,10]. However, realizing this potential requires a clear-eyed assessment of the technology's current limitations.

Indeed, the educational promise of these multimodal tools is accompanied by well-documented technical limitations that must be critically examined. First, at the perceptual level, generative models are prone to producing semantically or structurally inconsistent images, such as distorted objects, illogical spatial relationships, or garbled text, which could mislead learners or demand additional cognitive effort to interpret [11]. Second, at the socio-cultural level, these models often perpetuate and amplify biases inherent in their training data. For instance, they may systematically over-represent specific cultural archetypes when visualizing common concepts,

thereby reinforcing stereotypes rather than broadening cross-cultural understanding [12]. Third, at the practical level, inconsistencies in output quality and the evolving nature of synthetic media pose challenges for integration. Variations in color fidelity or detail can introduce unintended perceptual noise, while the fact that the detectability of AI-generated images changes over their online lifespan complicates authentication and responsible use [13]. Finally, unresolved copyright ambiguities arise when generated visuals inadvertently replicate elements of existing artworks, creating ethical dilemmas for educational dissemination.

Considering the limitations of conventional vocabulary teaching methods, this study explores a framework that incorporates AI-generated images into noun-focused multimodal learning. This approach mainly focus on the systematic creation and integration of images that are both semantically relevant and rich in perceptual detail. These visuals are designed to reflect meaningful contexts and include variations in texture, lighting, and perspective, helping learners form stronger associations between word form, meaning, and usage. The study contributes in four key areas: (1) proposing a structured method for generating targeted visual stimuli to support noun learning; (2) providing experimental evidence that such materials can improve vocabulary retention compared to traditional resources; (3) presenting a metric for evaluating the semantic consistency of visual aids used in instruction; and (4) offering practical insights into how AI-based tools can be used to design personalized, context-aware learning environments for vocabulary acquisition.

To address the pivotal question of how the mnemonic benefits of AI-generated visuals can be harnessed while mitigating their documented risks, this study adopts an experimental approach. By systematically comparing vocabulary learning outcomes under a text-only baseline and a multimodal condition employing AI-generated images, it seeks to provide empirical evidence on the net educational effect of this technology. The findings aim to move the discourse beyond theoretical debate, offering balanced insights for educators and designers on the practical integration of generative tools in language learning contexts.

## Related work

Vocabulary retention has long been a central topic in second language acquisition (SLA) and cognitive linguistics. Research has shown that nouns are more easily retained than other word classes due to their concreteness, imageability, and semantic centrality in sentence construction [4,9,14,15]. Nouns often serve as semantic anchors, facilitating mental imagery and contextual association, which are critical for long-term memory formation [16]. Studies in psycholinguistics also suggest that learners form stronger memory traces for concrete nouns than for abstract or functional words [10,17], supporting the rationale for focusing on noun retention in multimodal instruction [9]. This visual advantage makes nouns particularly suitable for multimodal learning environments, especially when paired with AI-generated imagery that reflects real-world contexts [10].

Multimodal approaches to vocabulary learning—incorporating text, image, audio, and video—have gained traction for their ability to engage multiple cognitive channels [5]. The Dual Coding Theory posits that information encoded both verbally and visually is more likely to be retained [10]. More recent work emphasizes semantic scene construction, where learners encounter vocabulary within meaningful, context-rich environments. However, traditional multimodal materials such as textbooks, films, and websites often suffer from semantic dispersion, where target words are embedded in noisy or irrelevant contexts, reducing instructional efficiency [18]. Recent discussions in multimodal learning also highlight the importance of material-based encoding, where the texture and physical properties of visual stimuli contribute to deeper semantic processing. Despite this, few studies have systematically examined how visual materiality—such as surface texture, color, and physical realism—affects vocabulary retention.

AI-generated Content has recently emerged as a transformative tool in educational technology [10,19]. Models such as DALL·E, Midjourney, and Stable Diffusion can generate high-resolution, context-specific images from textual prompts, enabling on-demand creation of instructional materials. In vocabulary learning, AI-generated Content offers the potential to produce semantically coherent, visually diverse, and pedagogically aligned stimuli tailored to individual words and learners

[8]. While AI-generated Content has been explored in creative writing, art education, and simulation-based training, its application in language learning and vocabulary instruction remains underexplored [20,21]. Preliminary studies have shown that AI-generated visuals can enhance learner engagement and comprehension [10], but few have investigated their impact on memory performance, especially in comparison to traditional materials. Furthermore, current AI-generated Content applications rarely incorporate semantic consistency metrics to evaluate the pedagogical relevance of generated content, nor do they systematically apply material-based encoding strategies [9]. These gaps highlight the need for a more structured and cognitively informed approach to integrating AI-generated images into vocabulary instruction—an area this study aims to advance.

## Methodology

To investigate how visual support influences vocabulary learning, this study employs a within-subjects design, focusing specifically on noun retention. Participants were exposed to two instructional conditions: one relying solely on traditional text-only materials, and another incorporating images generated through artificial intelligence to complement the vocabulary items. Each set of words was taught separately, with a 48-hour gap between sessions to reduce potential interference from prior learning. The research procedure includes several key components: selecting appropriate vocabulary, generating contextually relevant images using AI tools, designing instructional materials tailored to each condition, and implementing assessments to measure learning outcomes.

### A. Participants

A total of N = 40 eligible university students were recruited from the College of Urban Construction, Anhui Vocational College of Defense Technology, via course announcements and voluntary sign-up sheets. The sample consisted of 17 females and 23 males, aged 18–22 years (M = 19.2, SD = 1.1). All participants were native speakers of Chinese and had received formal English instruction for at least six years. Self-reported majors included big data & accounting (n = 20) and engineering cost management (n = 20). No participant had any diagnosed colour-vision deficiency or uncorrected visual impairment. Ethical approval for the study was obtained from the Institutional Review Board of Anhui Vocational College of Defense Technology (Approval No. CJ-2024–12003). Prior to participation, all individuals provided written informed consent. They were informed about the study procedures, their right to withdraw at any time, and the confidentiality of their data. The signed consent forms are securely stored by the research team.

### B. Vocabulary selection

To ensure semantic clarity and visual representability, 400 concrete and high-frequency English nouns were selected. These words cover 20 fields. Each field contains two control groups, with 10 words in each control group. In each field, one control group was used in the traditional learning(text-only) condition, and the other in the multimodal (AI-generated images) condition. The division was balanced in terms of word frequency, imageability, and semantic diversity to ensure comparability across conditions. Table 1 presents a representative sample of the vocabulary items used in each condition, along with the number of AI-generated images created for each word in the visual learning module.

### C. AI-generated visual stimuli

For the AI-generated images condition, visual stimuli were generated using a state-of-the-art text-to-image generation technology based on diffusion models. Each noun was used to generate multiple images that varied in:

• Materiality (e.g., glass, metal, fabric);

• Semantic context (e.g., "security" on uniforms, signage, websites);

• Visual perspective (e.g., close-up, environmental, functional use).

**Table 1. Sample vocabulary items by learning condition and visual stimuli coverage.**

| ID | Word | Learning Condition | Visual Stimuli Type | Image Count |
|----|------|--------------------|--------------------|-------------|
| 1 | security | Multimodal | AI-generated images | 12 |
| 2 | surveillance | Multimodal | AI-generated images | 12 |
| 3 | legislation | Multimodal | AI-generated images | 12 |
| 4 | infrastructure | Multimodal | AI-generated images | 12 |
| 5 | poverty | Multimodal | AI-generated images | 12 |
| 6 | unemployment | Multimodal | AI-generated images | 12 |
| 7 | crime | Multimodal | AI-generated images | 12 |
| 8 | disaster | Multimodal | AI-generated images | 12 |
| 9 | refugee | Multimodal | AI-generated images | 12 |
| 10 | welfare | Multimodal | AI-generated images | 12 |
| 11 | community | Traditional | None (text only) | 0 |
| 12 | hierarchy | Traditional | None (text only) | 0 |
| 13 | regulation | Traditional | None (text only) | 0 |
| 14 | facility | Traditional | None (text only) | 0 |
| 15 | inequality | Traditional | None (text only) | 0 |
| 16 | redundancy | Traditional | None (text only) | 0 |
| 17 | violence | Traditional | None (text only) | 0 |
| 18 | calamity | Traditional | None (text only) | 0 |
| 19 | asylum | Traditional | None (text only) | 0 |
| 20 | subsidy | Traditional | None (text only) | 0 |

Prompt templates were carefully crafted to ensure semantic consistency and contextual relevance. All images were manually reviewed for accuracy and realism. Table 2 provides examples of the prompt templates used for selected vocabulary items. These templates reflect the diversity of contexts and visual features incorporated into the AI-generated images.

To visually demonstrate the diversity and realism of the generated content, Fig 1 presents a selection of nine representative images produced using the prompt templates listed in Table II. These examples illustrate variations in materiality, semantic context, and visual perspective for the word "security," highlighting the richness and flexibility of AI-generated stimuli used in the experiment. In addition, the differences in visual angle, texture, and contextual detail across the images encourage learners to engage in exploratory viewing and semantic verification. This type of visual engagement attracts attention and stimulates active cognitive processing, which in turn enhances encoding and strengthens memory formation. The perceptual richness and contextual variation embedded in the images enhance learners' interaction with the vocabulary, making the learning experience more immersive and memorable.

## D. Instructional design

All participants (N = 40) completed two learning sessions:

1. *Session 1 (Traditional Condition):* Learners studied 200 nouns(20 groups, one group every week) using text-only materials, including word, phonetic transcription, definition, and example sentence. No images were provided.

2. *Session 2 (Multimodal Condition):* Learners studied a different set of 200 nouns(20 groups, one group every week) using a structured, interactive system based on AI-generated images. Each word was taught through three progressive stages:

**Table 2. Prompt template examples for AI-generated image generation.**

| Word | Prompt template example |
|---|---|
| security | A close-up of a security guard's uniform with red embroidery reading 'Security.' |
| security | There is a Security office with a very clear "Security" sign on the wall. |
| security | Ultra-realistic 8K photo of a confident female security guard in a black tactical uniform with bright white "SECURITY" printed on her chest and back, standing at the marble lobby of a luxury high-rise, golden hour sunlight streaming through glass walls, shallow depth of field, Canon EOS R5 look. |
| security | Corporate minimalism: overhead symmetrical shot of a sleek white reception desk in a bright open-plan office, frosted glass door with subtle back-lit "SECURITY" lettering, two smartly dressed workers blurred in background, Hasselblad clean aesthetic. |
| security | Concert entrance, night: LED wristband scanner, young guard wearing reflective vest with glowing red "SECURITY", neon stage light bokeh behind, shallow depth, cinematic 50 mm lens. |
| security | Subway security checkpoint, Beijing Line 4: overhead daylight, bright yellow safety barrier with bold black "SECURITY" stenciling, uniformed guard waving handheld metal detector, commuters queued behind, 35 mm street-photography look, slight motion blur. |
| security | Minimalist flat-lay: matte black swipe card, silver lanyard, bold white sans-serif "SECURITY" printed on card, two crossed metallic pens, clean white background, top-down, soft shadows, Apple-style photography. |
| security | giant brushed-steel wall with the word "Security" in sleek sans-serif letters, soft natural light, Scandinavian minimalism, shallow depth of field, high-end architectural photography. |

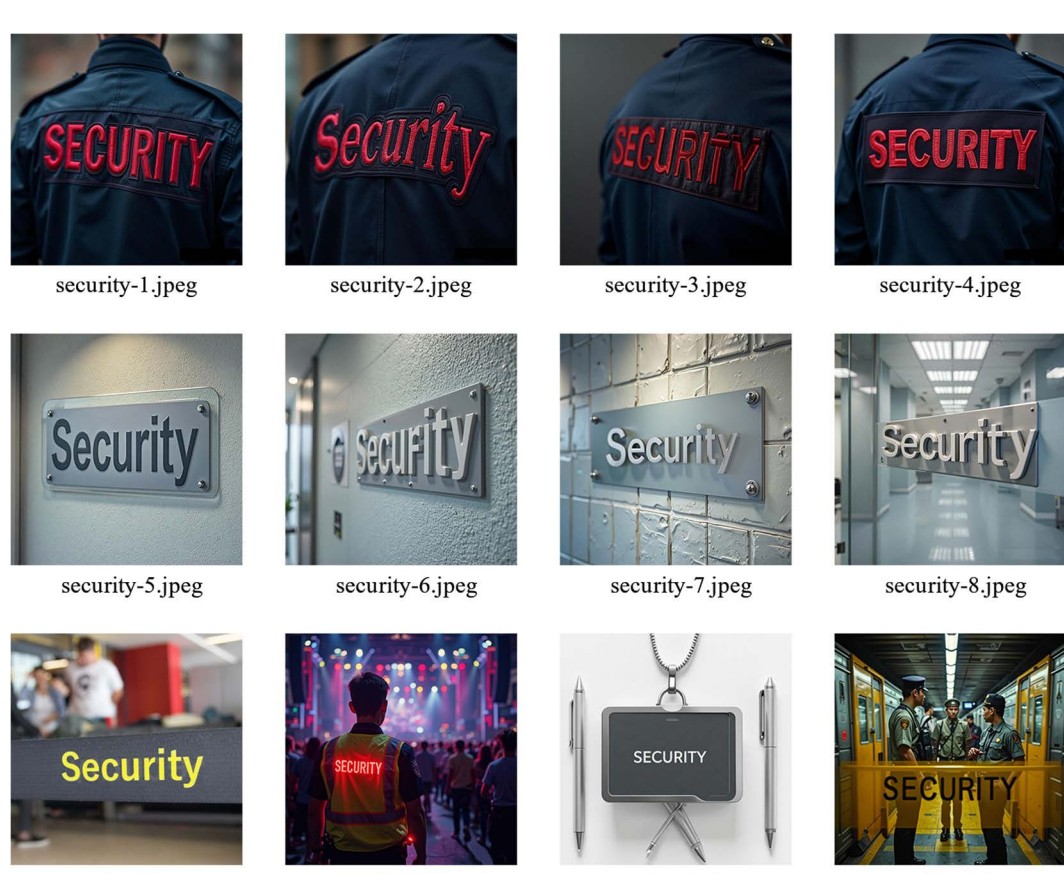

security-1.jpeg security-2.jpeg security-3.jpeg security-4.jpeg

security-5.jpeg security-6.jpeg security-7.jpeg security-8.jpeg

security-9.jpeg security-10.jpeg security-11.jpeg security-12.jpeg

**Fig 1. AI-generated visual stimuli for the word "Security".** A selection of nine representative images produced using the prompt templates listed in Table II, illustrating variations in materiality, semantic context, and visual perspective. Images were generated using generative AI services: specifically, Stable Diffusion (Stability AI's text-to-image model), Doubao (ByteDance's text-to-image model), and SenseMirage (SenseTime's text-to-image model).

a. Basic Learning: Learners were presented with the target word, its pronunciation, and an initial AI-generated image designed to support basic recognition. Once the learner felt the word was memorized, they could manually proceed to the next stage.

b. Semantic Expansion: Learners continued studying the same word, accompanied by a series of AI-generated images (typically five or more), each emphasizing different materials, contexts, or perspectives. Images were displayed one at a time, and learners could click to view the next image at their own pace, allowing for individualized semantic exploration.

c. Contextual Application: Learners engaged with additional AI-generated images paired with simple contextual sentences (typically three), illustrating the word's use in real-world scenarios. As in the previous stage, image and sentence progression was learner-controlled, enabling flexible and personalized semantic generalization.

To illustrate the instructional flow of the multimodal module, Fig 2 presents a schematic diagram of the three-stage learning process. This visual representation highlights the structured progression from basic recognition to semantic generalization and contextual application.

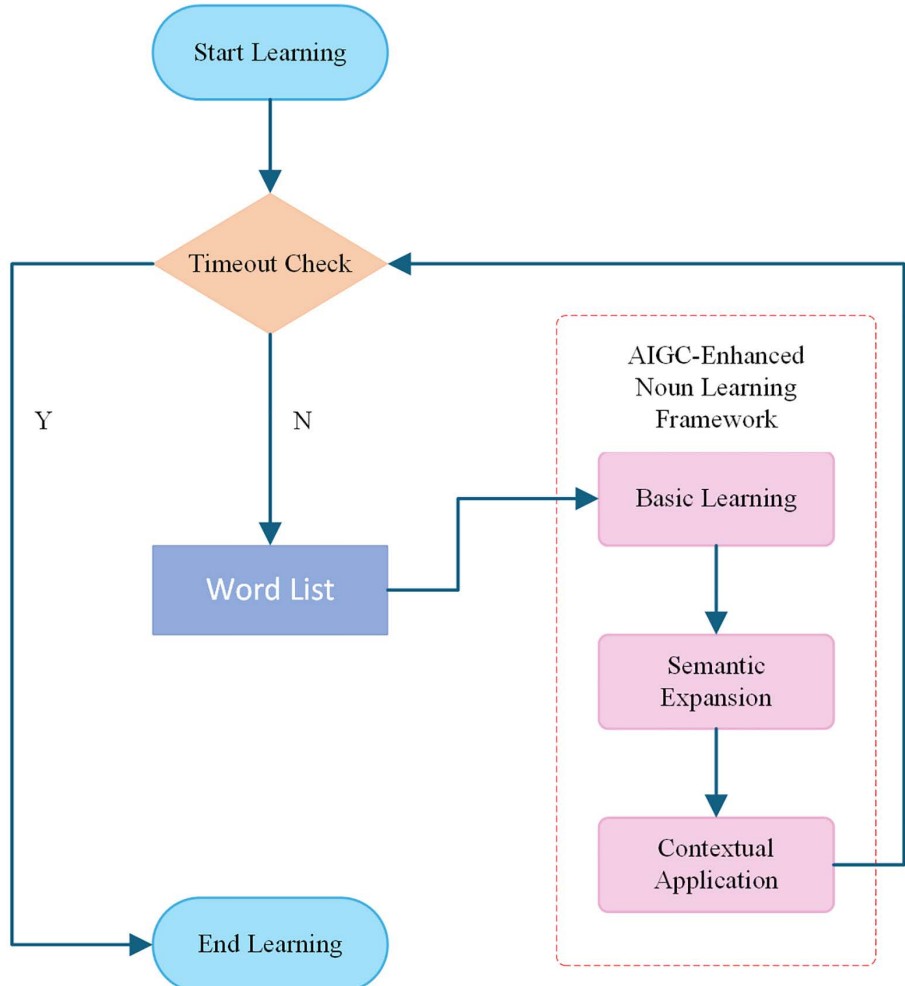

**Fig 2. Instructional flow of multimodal vocabulary learning module.**

## E. Assessment procedures

To evaluate learners' retention and semantic understanding of vocabulary, assessments were conducted at two distinct time points:

- Immediate Recall Phase: administered directly after each learning session;
- Delayed Recall Phase: conducted 48 hours after the initial learning session.

At both time points, participants completed the same set of three tasks, each designed to tap into different cognitive aspects of vocabulary knowledge:

- Definition Selection Task – a multiple-choice format assessing recognition of word meanings;
- Definition Writing Task – requiring learners to produce definitions, evaluating productive semantic recall;
- Image-to-Word Matching Task – a transfer task designed to probe the quality and accessibility of the mental representations formed during learning. It required participants to access their semantic understanding of a noun and identify its corresponding visual referent among distractors. This task does not assume prior image exposure; rather, it assesses the robustness of the conceptual link between a word and its meaning, which is theoretically strengthened through multimodal encoding.

This structure allowed for a comprehensive comparison of learning outcomes across instructional conditions at the levels of recognition, production, and cross-modal transfer, while also capturing both immediate and sustained retention effects.

## F. Data analysis

In order to compare the effectiveness of traditional text-only instruction and multimodal instruction with AI-generated images, this study applied statistical methods suitable for a within-subjects design. Such methods mainly used paired samples t-tests and repeated measures ANOVA. And the analyses were used to examine differences in learner performance under different instructional conditions and between two time points: immediately after learning and 48 hours later.

### Retention rate calculation

To better capture how well participants retained vocabulary over time, a retention rate was calculated for each individual. This metric reflects the proportion of correctly recalled items in the delayed test relative to the immediate test, as shown in (1). By using this approach, the study was able to assess not only initial learning outcomes but also the durability of memory across conditions.

$$R_i = \frac{C_i}{T} \tag{1}$$

Where,
$R_i$ = retention rate of participant $i$
$C_i$ = number of correctly recalled words by participant $i$
T = total number of words learned in that condition (i.e., 20)
Group-level retention rates were then averaged and compared across conditions.

### Semantic association rating

The learners' semantic generalization ability was assessed by the mean semantic rating score, which was calculated using (2), based on a 5-point Likert scale.

$$\overline{S} = \frac{1}{n} \sum_{i=1}^{n} S_i$$

(2)

Where,

$S_i$ = rating score given by participant $i$

n = total number of participants

Standard deviation (SD) was also reported to assess variability

### Definition accuracy and matching tasks

Accuracy scores were computed using (3), representing the proportion of correct responses in each task, so as to evaluate task-specific performance.

$$A = \frac{C}{N}$$

(3)

Where,

A = accuracy score

C = number of correct responses

N = total number of items

These scores were compared across conditions using paired t-tests to determine whether Multimodal instruction led to significantly higher performance.

### Statistical testing

a. Paired Samples t-Test was used to compare mean scores between the traditional and multimodal (AI-generated images) conditions for each task.

b. Repeated Measures ANOVA was applied to examine interaction effects between instructional condition and time (immediate vs. delayed).

c. Effect sizes (Cohen's d) were reported to quantify the magnitude of differences.

## Results

This chapter presents the core findings of the experiment, which compared vocabulary learning outcomes under two instructional modes: traditional text-based instruction and multimodal instruction with AI-generated images. The design evaluates the benefit of adding visual information in this context, rather than isolating the specific contribution of the image-generation method. The analysis focuses on how this modal difference influenced noun retention, particularly in terms of recall accuracy, semantic depth, and task performance. Results are organized by assessment type and time point, examining how the multimodal presentation supported learners in forming stronger and more lasting word associations.

### A. Memory retention performance

Participants completed both immediate and delayed recall tests after each learning session. Retention rates were calculated as defined in (1), and the results indicate that the multimodal condition consistently outperformed the traditional condition.

The multimodal group achieved an average immediate recall rate of 0.85 and a delayed recall rate of 0.75, while the traditional group scored 0.65 and 0.45, respectively. Paired samples t-tests revealed statistically significant differences ($p < 0.01$), with a large effect size (Cohen's $d = 0.82$). To further illustrate the impact of vocabulary load on memory retention, Fig 3 presents a line chart comparing recall rates across different word counts for both groups.

Fig 4 illustrates recall performance across increasing vocabulary loads under two instructional conditions. It's obvious that the overall recall rates drop as the number of target nouns increases under both conditions. However, the multimodal method keeps outperforming traditional text-only instruction. It turns out that even after multiple learning sessions within a set time, learners who were exposed to AI-generated images remembered more words. This result suggests that rich semantic contexts and specific visual cues help strengthen word-meaning connections, making noun vocabulary easier to remember and less likely to be forgotten over time.

## B. Task-specific performance

To explore how learners processed vocabulary beyond simple recall, two tasks were used: definition selection and image-to-word matching. These tasks were designed to assess not just whether a word was remembered, but the quality of semantic understanding and the strength of cross-modal connections in mental representation.

Results showed a clear advantage for the multimodal condition in the depth of semantic processing. In the definition selection task, the average accuracy was 0.80 for the multimodal group, compared to 0.60 in the traditional group. More critically, in the image-to-word matching task—a measure of cross-modal transfer—the performance gap was substantial (0.85 vs. 0.45). This stark contrast indicates that the multimodal input enabled learners to form semantic representations that were not only more robust but also more readily accessible and applicable when meaning needed to be retrieved and mapped to a novel visual context. These differences remained consistent even after multiple learning rounds within a fixed time frame, suggesting that integrated visual-textual encoding supported the construction of more stable and functionally flexible conceptual representations.

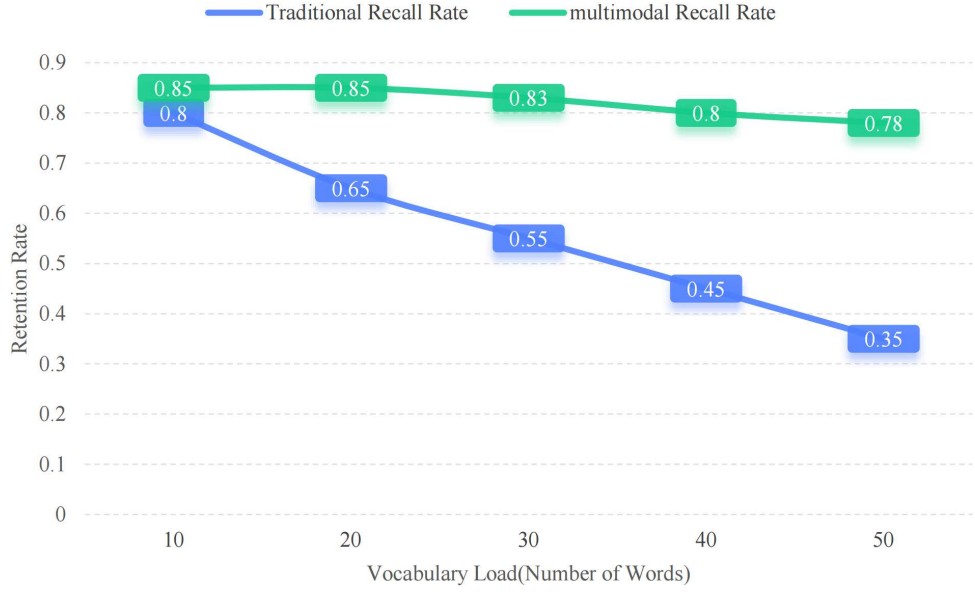

**Fig 3. Retention curve: Immediate vs. Delayed recall.**

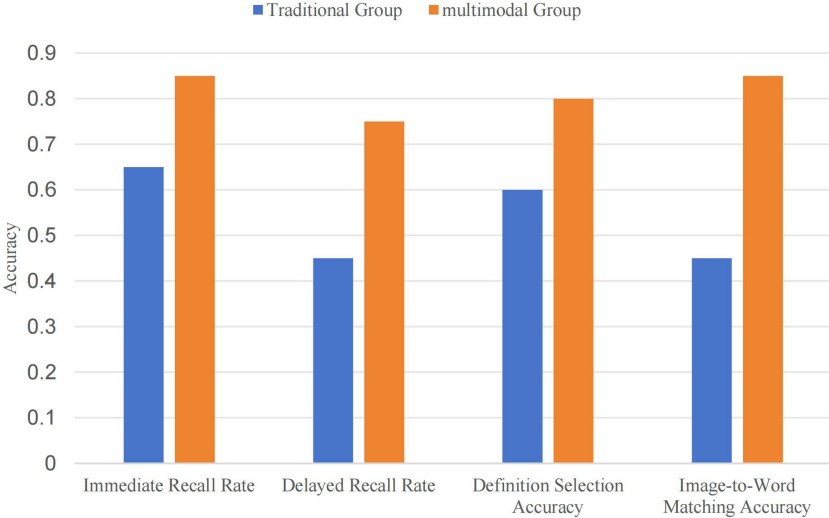

**Fig 4. Retention rate vs. Vocabulary load: Traditional vs. Multimodal method.**

Statistical analysis supported these findings, with large effect sizes observed (Cohen's $d = 0.75$ for definition selection, $d = 0.70$ for image-to-word matching). The results reinforce the idea that multimodal instruction enhances vocabulary learning by fostering richer, more interconnected mental representations that facilitate both recognition and cross-modal application.

## C. Semantic evaluation

Finally, to make clear how well the learners internalized the word meanings, participants rated each target noun on a 5-point scale, based on how clearly they thought they understood its meaning. This subjective measure was used to complement task-based results and reflect learners' semantic impressions.

The ratings of the tests revealed a noticeable gap between the two instructional methods. Learners in the multimodal group gave an average rating of 4.1, while those in the traditional group averaged 3.2. Even after multiple learning rounds within the same time frame, the multimodal group were still more impressed by the testing words, from which we can conclude that the AI-generated images helped the learners form more vivid and coherent mental representations of the target nouns.

Statistical tests confirmed the reliability of this difference ($p < 0.01$), with a large effect size (Cohen's $d = 0.88$). These findings support the idea that multimodal input, with meaningful visual scenes, can deepen learners' semantic understanding and improve long-term retention.

## D. Summary

The results show that multimodal instruction with AI-generated images led to more effective noun learning across multiple aspects. Learners recalled more words, performed better on meaning-based tasks, and reported clearer semantic impressions. Even under the same time constraints and after repeated learning sessions, the group exposed to AI-generated imagery consistently showed stronger retention and understanding. These findings highlight the value of pairing visual detail with lexical semantics, especially in multimodal environments focused on concrete vocabulary.

## Discussion

For a large number of English learners, memorizing English nouns is mainly about repetition to resist the Ebbinghaus forgetting curve. Most of them seldom notice that vocabulary learning also depends on how well the words are anchored in context and perception. And such anchoring process needs to build connections between the word form, its semantic role, and how it appears or functions in real-world situations and things like that.

From the study we can see that learners who used AI-generated images to remember nouns showed stronger retention and clearer understanding of the target nouns. This was not only due to the presence of semantic scenes, but also because of the visual details embedded in those scenes—such as object shape, spatial arrangement, interaction cues, etc. These elements provided multiple entry points for memory encoding, allowing the learners to associate the words with both the semantic meaning and their physical appearance.

Take the word crane for example, it was not simply defined or placed in a port setting in the test, it was visually distinguished by its structure and function, thus helped the learners differentiate it from similar nouns like tower or container. According to the test, such visual precision, combined with contextual relevance, appears to support deeper semantic processing and more durable memory traces.

These findings reveal that, when carefully designed to reflect both semantic context and visual specificity, multimodal instruction can significantly promote vocabulary learning, especially for concrete nouns, because the learning of nouns is comparatively more beneficial for their rich perceptual input.

## Conclusion and implications

This study explored how multimodal presentation can support second language learners in acquiring English nouns. Compared with traditional text-only instruction, the multimodal condition led to better recall, clearer semantic understanding, and stronger performance in meaning-based tasks.

Learners benefited not only from seeing words in context, but also from the visual details that helped distinguish and anchor meanings. These findings suggest that pairing verbal input with rich, targeted imagery can make vocabulary more memorable—especially when learners are dealing with unfamiliar or abstract nouns.

### A. Pedagogical implications

In terms of classroom teaching, these results can support the integration of AI-generated visuals into vocabulary instruction. Teachers can leverage generative tools to produce tailored images that capture the semantic traits of target words, aiding learners in forming stronger connections between word form and meaning. This method is especially beneficial in environments where learners have limited exposure to real-world language usage or face difficulties with abstract vocabulary, such as various terms in different industries.

### B. Research implications

Future studies could extend this work to other word types, such as verbs, adjectives etc., or to idiomatic expressions, and explore how visual support affects long-term retention. Qualitative methods like learner interviews may also help uncover how learners process and internalize multimodal input. Additionally, cross-linguistic studies could examine whether learners from different language backgrounds respond differently to visual support and what their differences are.

### C. Practical applications

The findings also hold potential for the application in digital learning settings. Since AI-based image-generation tools can be used to generate visuals that are much richer in context to improve the effectiveness of vocabulary learning, they can be applied by various language learning platforms, mobile applications, intelligent tutoring systems, etc. to automatically

produce images that convey both the word's meaning and its typical usage scenarios. In this way, language learners will have better learning results which is both effective and durable as well as better learning experience which is more interactive and personalized.

## Author contributions

**Conceptualization:** Gaojie Ye.

**Data curation:** Shibo Yan.

**Formal analysis:** Shibo Yan.

**Investigation:** Gaojie Ye.

**Methodology:** Gaojie Ye.

**Project administration:** Gaojie Ye.

**Software:** Shibo Yan.

**Supervision:** Gaojie Ye.

**Writing – original draft:** Gaojie Ye, Shibo Yan.

**Writing – review & editing:** Gaojie Ye, Shibo Yan.

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
