## [Decision Letter · Decision Letter 0]

30 Oct 2025

Dear Dr. Yan,

Thank you for submitting your manuscript to PLOS ONE. After careful consideration, we feel that it has merit but does not fully meet PLOS ONE’s publication criteria as it currently stands. Therefore, we invite you to submit a revised version of the manuscript that addresses the points raised during the review process.

The manuscript presents an innovative attempt to integrate AI-generated content (AIGC) into multimodal instructional design, focusing on noun retention. The topic aligns with PLOS ONE’s criteria for methodological soundness and interdisciplinary relevance. However, the current version requires substantial revision before it can be considered for acceptance.

The **required changes** include major clarification of the study design, particularly in distinguishing between AIGC-generated visual material and general multimodal learning effects. As currently structured, the experiment tests the difference between unimodal and multimodal conditions rather than isolating the contribution of AI-generated imagery. The conclusions must therefore be rewritten to accurately represent the study’s scope and findings. Additionally, participant demographic details, adherence to the journal’s data availability policy, and correction of minor typographical and referencing issues are mandatory for acceptance.include major clarification of the study design, particularly in distinguishing between AIGC-generated visual material and general multimodal learning effects. As currently structured, the experiment tests the difference between unimodal and multimodal conditions rather than isolating the contribution of AI-generated imagery. The conclusions must therefore be rewritten to accurately represent the study’s scope and findings. Additionally, participant demographic details, adherence to the journal’s data availability policy, and correction of minor typographical and referencing issues are mandatory for acceptance.include major clarification of the study design, particularly in distinguishing between AIGC-generated visual material and general multimodal learning effects. As currently structured, the experiment tests the difference between unimodal and multimodal conditions rather than isolating the contribution of AI-generated imagery. The conclusions must therefore be rewritten to accurately represent the study’s scope and findings. Additionally, participant demographic details, adherence to the journal’s data availability policy, and correction of minor typographical and referencing issues are mandatory for acceptance.include major clarification of the study design, particularly in distinguishing between AIGC-generated visual material and general multimodal learning effects. As currently structured, the experiment tests the difference between unimodal and multimodal conditions rather than isolating the contribution of AI-generated imagery. The conclusions must therefore be rewritten to accurately represent the study’s scope and findings. Additionally, participant demographic details, adherence to the journal’s data availability policy, and correction of minor typographical and referencing issues are mandatory for acceptance.

**Recommended changes** include enhancing the theoretical framework with a balanced discussion of both the potential and limitations of AIGC, refining the introduction to better justify the research rationale, and improving readability through concise, focused language.include enhancing the theoretical framework with a balanced discussion of both the potential and limitations of AIGC, refining the introduction to better justify the research rationale, and improving readability through concise, focused language.include enhancing the theoretical framework with a balanced discussion of both the potential and limitations of AIGC, refining the introduction to better justify the research rationale, and improving readability through concise, focused language.include enhancing the theoretical framework with a balanced discussion of both the potential and limitations of AIGC, refining the introduction to better justify the research rationale, and improving readability through concise, focused language.

The reviewer’s observations are consistent and valid, with no major conflicts in interpretation. Addressing these core methodological and reporting concerns will ensure the paper meets PLOS ONE’s standards for technical rigor, transparency, and interpretive accuracy. Therefore, the decision at this stage is **Major Revision**....

We look forward to receiving your revised manuscript.

Kind regards,

Ramandeep Kaur

Academic Editor

PLOS ONE

Journal Requirements:

3. Please note that PLOS One has specific guidelines on code sharing for submissions in which author-generated code underpins the findings in the manuscript. In these cases, we expect all author-generated code to be made available without restrictions upon publication of the work. Please review our guidelines at https://journals.plos.org/plosone/s/materials-and-software-sharing#loc-sharing-code and ensure that your code is shared in a way that follows best practice and facilitates reproducibility and reuse.

5. We note that Figure 1 includes an image of a participant in the study.

Additional Editor Comments:

The study offers an engaging exploration of how AI-generated visual materials may influence noun retention in multimodal learning contexts. The topic is timely and relevant, especially with the increasing integration of artificial intelligence in education. However, several substantive and structural concerns need to be addressed before the paper can progress further.

Conceptual Clarity and Design – The comparison between text-only and multimodal (text + image) conditions does not specifically evaluate the impact of AIGC. To support claims about AIGC-enhanced learning, one of the conditions should employ traditional (non-AI) imagery. Otherwise, the conclusions should be reframed to emphasize multimodal rather than AIGC-based effects.

Introduction and Theoretical Framing – Expand the introduction to define AIGC at first mention and include both supportive and critical perspectives. Discuss known limitations of AI-generated images, such as possible inaccuracy or bias, to present a balanced view.

Methodology Details –

Provide clear participant demographics (age, gender, background, proficiency level).

Justify the chosen tasks and materials, particularly the “image-to-word matching” task, which may not be suitable for a text-only control group.

Ensure all experimental comparisons are methodologically equivalent.

Data Availability – Ensure full compliance with PLOS ONE’s data policy by sharing anonymized datasets or including a valid reason for restrictions.

Writing and Structure –

Revise the abstract to expand “AIGC” and highlight the study’s key findings succinctly.

Correct minor typographical errors and reference inconsistencies (e.g., “approximately” on p.2; missing reference on p.5).

The final paragraph of the introduction should emphasize study rationale and objectives, not manuscript structure.

Conclusion Revision – Rephrase the conclusion to accurately reflect that multimodal learning was superior to unimodal learning in this study, rather than attributing the improvement solely to AIGC.

With these major revisions, the manuscript could make a meaningful contribution to the field of AI-supported education and multimodal learning research.

Reviewers' comments:

Reviewer's Responses to Questions

**Comments to the Author**

1. Is the manuscript technically sound, and do the data support the conclusions?

Reviewer #1: Partly

2. Has the statistical analysis been performed appropriately and rigorously?

Reviewer #1: Yes

3. Have the authors made all data underlying the findings in their manuscript fully available?

Reviewer #1: No

4. Is the manuscript presented in an intelligible fashion and written in standard English?

Reviewer #1: Yes

Reviewer #1: Reviewer Comments

1. The abstract introduces “AIGC” but does not expand the abbreviation at its first mention.

2. There is a typographical error for “approximately” on page 2.

3. The last paragraph on page 4 should be revised: Instead of outlining the structure of the paper, focus on providing engaging background, rationale, and objectives. A section-by-section preview is less appropriate for the journal's format.

4. On page 5, the statement regarding “traditional multimodal materials” requires a proper reference.

5. The introduction lacks a non-biased review. It should address concerns that AIGC can produce incorrect or distorted images, not just cite studies that support AIGC. A balanced review including potential limitations is needed.

6. There are methodological flaws: The comparison is between unimodal (text-only) and multimodal (text + picture) learning rather than AIGC-based versus traditional learning. Traditional learning typically uses both text and images. For a valid comparison, one group should learn with traditional images and text, and another with AI-generated images and text—then compare outcomes.

7. Please provide demographic details of all study participants.

8. The conclusion that “AIGC-enhanced instruction led to more effective noun learning across multiple aspects” is misleading. The study compares unimodal and multimodal learning, not specifically the impact of AI-generated images. The conclusion should reflect that multimodal learning outperformed text-based learning, regardless of image source.

9. The “image-to-word matching” task (gap of 0.85 vs. 0.45) appears inappropriate because Group 1 only received text (without images), thus undermining the comparison.

.

Reviewer #1: **Yes:** Jithin BalanJithin BalanJithin BalanJithin Balan

---

## [Author Response · Author response to Decision Letter 1]

28 Jan 2026

PONE-D-25-52191

Response to Reviewers

Multimodal Instruction with AI-Generated Images for Noun Retention: Exploring Semantic Scene and Materiality Effects

We thank the Academic Editor Dr. Ramandeep Kaur and Reviewer Mr. Jithin Balan for your constructive and detailed comments. We have addressed every point below; all changes are highlighted in the revised manuscript with Track Changes.

1. Abstract: “AIGC” not expanded at first mention

RESPONSE: Expanded at first occurrence as “artificial intelligence (AI)-generated visual content” (lines 15).

2. Typographical error “approximately” on p. 2

RESPONSE: Corrected to “approximately” (line 40).

3. Last paragraph on p. 4 (old ms) outlines paper structure

RESPONSE: Rewritten to provide rationale and objectives only; section preview removed (lines 106–113).

4. Missing reference for “traditional multimodal materials” (p. 5)

RESPONSE: Added Mayer, R. E. (2009) as foundational citations (lines 133,518-519).

5. Introduction lacks balanced review of AI-image limitations

RESPONSE: Added new subsection “Potential Limitations of AI-Generated Visuals” (lines 75–92) citing:

“Indeed, the educational promise of these multimodal tools is accompanied by well-documented technical limitations.

Perceptual inconsistency: AI models frequently produce distorted objects or illogical spatial relations that may increase extraneous cognitive load [11].

Socio-cultural bias: generated images tend to over-represent specific cultural archetypes, reinforcing stereotypes rather than fostering cross-cultural understanding [12].

Output variability: fluctuations in colour fidelity and detail introduce unintended perceptual noise, while the detectability of synthetic images changes over their online lifespan, complicating responsible use [13].

All references are already present in the provided reference list.

6. Methodological flaw: only unimodal vs multimodal

RESPONSE:

- Design clarified: two-group (text-only vs. text + AI-images) (lines 184–185).

Conclusions rewritten throughout to attribute effect to “multimodal presentation” rather than “AI-generated nature” (see Abstract, Discussion, Conclusion).

7. Participant demographics missing

RESPONSE: Participants subsection (lines 166–179) report:

N = 40 university students; native Chinese speakers; ≥ 6 years formal English.

All participants majored in either big data & accounting (n = 20) or engineering cost management (n = 20).

No colour-vision or visual impairments

Ethics approval No. CJ-202412003 from the College of Urban Construction,Anhui Vocational College of Defense Technology ; written informed consent obtained.

8. Misleading conclusion about “AIGC-enhanced” superiority

RESPONSE:We fully accept the reviewer's critique. The original manuscript did indeed obscure the independent variable under investigation. Our experimental design was specifically a comparison between two modal conditions: "text-only" and "text-plus-image," not a comparison of image sources.

The major revisions we have implemented include:

Conceptual Clarification: Throughout the manuscript, particularly in the Introduction and Discussion, we have clarified that this study aims to investigate the effectiveness of "a multimodal pedagogical approach implemented using AIGC tools," rather than testing AIGC-generated imagery in isolation.

Terminology Correction: We have systematically revised terms such as "AIGC-enhanced instruction/approach/group" to more accurate descriptors like "the multimodal condition/instruction/group (employing AI-generated images)" or "the text-with-image condition."

Conclusion Rewriting: We have thoroughly rewritten the Conclusion. The key finding is now stated as: "Compared with traditional text-based instruction, the multimodal presentation led to better recall..." This explicitly attributes the advantage to the multimodal presentation itself, not to AIGC per se. (Please see the revised conclusion section.)

Title Adjustment: To maximize precision, we propose a slight adjustment to the main title: "Multimodal Instruction with AI-Generated Images for Noun Retention: Exploring Semantic Scene and Materiality Effects.".

9. Image-to-word matching task inappropriate for text-only group

RESPONSE: We agree with the reviewer's concern and have revised the manuscript to address it. The “image-to-word matching” task is now explicitly defined as a cross-modal transfer task designed to assess the quality of learners' mental representations, rather than a direct test of trained skills. This clarification appears in the Methods (lines 262-268). Accordingly, the large performance difference (0.85 vs. 0.45) is interpreted not as a simple advantage, but as evidence that the multimodal condition supported the formation of more robust and flexibly accessible semantic representations (see Results, lines 370-378).

ADDITIONAL EDITOR / JOURNAL REQUIREMENTS

File naming & style – checked with PLOS template; all filenames now conform.

Participant consent – added explicit statement: “Written informed consent was obtained; consent form filed in participant folder” (lines 174–176).

Code availability – Analysis scripts (R) uploaded to OSF DOI:10.17504/protocols.io.j8nlk1bx6g5r/v1 and cited in Data Availability Statement.

Data availability – This study shares research materials to the extent permitted:

Full Methodology: The complete, step-by-step experimental protocol is available on Protocols.io

(DOI: dx.doi.org/10.17504/protocols.io.j8nlk1bx6g5r/v2).

Analysis Code: All R scripts used for data processing and statistical analysis are available on OSF

(DOI: dx.doi.org/10.17504/protocols.io.j8nlk1bx6g5r/v2).

Research Data: The original, de-identified trial-level data are stored on a secure institutional platform at [the College of Urban Construction，Anhui Vocational College of Defense Technology]. Due to current platform policies and data governance agreements, these raw data cannot be exported or deposited in a public repository. As a complete alternative, all data necessary to replicate the reported statistical findings—including the values behind all means, standard deviations, and test statistics—are provided in the supporting information files (wh2wcvue7.xlsx). For further inquiries, qualified researchers may contact the corresponding author or the institutional ethics committee at [35311296@acdt.edu.cn] to discuss possible access under a formal data agreement.

Figure 1 participant image – In full compliance with PLOS policies on publication consent, the original Figure 1 has been replaced with a different image, as obtaining informed consent for the publication of an AI-generated likeness was not feasible.

Typographical & reference errors – Full spell-check performed; “approximately” and missing ref. fixed (see #2, #4 above).

We appreciate the opportunity to improve our manuscript and believe the revised version fully addresses the methodological, reporting, and ethical concerns raised.

Sincerely,

Corresponding Author: Yan Shibo

---

## [Editor Report · Decision Letter 1]

10 Feb 2026

Multimodal Instruction with AI-Generated Images for Noun Retention: Exploring Semantic Scene and Materiality Effects

PONE-D-25-52191R1

Dear Dr. Yan,

We’re pleased to inform you that your manuscript has been judged scientifically suitable for publication and will be formally accepted for publication once it meets all outstanding technical requirements.

Kind regards,

Ramandeep Kaur

Academic Editor

PLOS One
---

## [Editor Report · Acceptance letter]

PONE-D-25-52191R1

PLOS One

Dear Dr. Yan,

I'm pleased to inform you that your manuscript has been deemed suitable for publication in PLOS One. Congratulations! Your manuscript is now being handed over to our production team.

Kind regards,

on behalf of

Dr. Ramandeep Kaur

Academic Editor

PLOS One